# *rab-27* acts in an intestinal pathway to inhibit axon regeneration in *C. elegans*

**Alexander T. Lin-Moore** [1], **Motunrayo J. Oyeyemi** [2], **Marc Hammarlund** [1,3]*

**1** Department of Genetics, Yale University School of Medicine, New Haven, Connecticut, United States of America, **2** Yale College, New Haven, Connecticut, United States of America, **3** Department of Neuroscience, Yale University School of Medicine, New Haven, Connecticut, United States of America

* marc.hammarlund@yale.edu

## Abstract

Injured axons must regenerate to restore nervous system function, and regeneration is regulated in part by external factors from non-neuronal tissues. Many of these extrinsic factors act in the immediate cellular environment of the axon to promote or restrict regeneration, but the existence of long-distance signals regulating axon regeneration has not been clear. Here we show that the Rab GTPase *rab-27* inhibits regeneration of GABAergic motor neurons in *C. elegans* through activity in the intestine. Re-expression of RAB-27, but not the closely related RAB-3, in the intestine of *rab-27* mutant animals is sufficient to rescue normal regeneration. Several additional components of an intestinal neuropeptide secretion pathway also inhibit axon regeneration, including NPDC1/*cab-1*, SNAP25/*aex-4*, KPC3/*aex-5*, and the neuropeptide NLP-40, and re-expression of these genes in the intestine of mutant animals is sufficient to restore normal regeneration success. Additionally, NPDC1/*cab-1* and SNAP25/*aex-4* genetically interact with *rab-27* in the context of axon regeneration inhibition. Together these data indicate that RAB-27-dependent neuropeptide secretion from the intestine inhibits axon regeneration, and point to distal tissues as potent extrinsic regulators of regeneration.

## Author summary

Since most neurons are not replaced over an organism's lifetime, neurons must regenerate damaged axons in order to restore function after injury. Despite the importance of regeneration to organism function, behavior and survival, regeneration is often actively inhibited in mature animals. Our results show that distant tissues can block regeneration. We show for the first time that the intestine secretes factors that inhibit axon regeneration, and that blocking this pathway improves regeneration.

## Introduction

Unlike many other tissues, where cells respond to injury through proliferation and replacement, cells in the nervous system are not usually replaced following axon damage. Instead,

index.htm) grants (R01 NS098817 and R01 NS094219) to M.H. The funders had no role in study design, data collection and analysis, decision to publish, or preparation of the manuscript.

**Competing interests:** The authors have declared that no competing interests exist.

neurons rely on axon regeneration to restore the connectivity necessary for function. Despite its importance, however, axon regeneration is often inhibited in vivo, leading to permanent loss of nervous system function after injury.

A neuron's axon regeneration capacity is extensively regulated by contacts with the extracellular environment of the injured axon. In the mammalian central nervous system, myelin-associated transmembrane signals Nogo, MAG and OMgp potently inhibit post-injury growth through direct interaction with neuronal receptors like Ngr1 and PTPσ [1,2]. In *C. elegans*, which lacks myelin-associated regeneration inhibitors, the peroxidasin PXN-2 and syndecan (SDN-1) control the integrity and signaling topography of the extracellular matrix to negatively or positively regulate regeneration success, respectively [3,4]. Thus, a neuron's local environment and neighbor cells influence its regenerative capacity.

In addition to responding to their local environment and neighbors, neurons respond to secreted, long-range signals from distant tissues, which can regulate neuronal programs ranging from synapse patterning to complex behaviors [5–7]. But for axon regeneration, the existence of long-range inhibitory signals in vivo has not been clear. We have previously identified the Rab GTPase *rab-27* as a conserved inhibitor of axon regeneration [8]. Here we show that *rab-27* inhibits regeneration of D-type motor neurons in *C. elegans* through activity in the intestine. We further show that inhibition of axon regeneration involves an intestinal secretory pathway involved in neuropeptide secretion. Together these results indicate that the *C. elegans* intestine inhibits axon regeneration, and point to long-distance, extrinsic signaling as a novel mechanism of axon regeneration regulation.

## Results

### An intestinal function for RAB-27 in axon regeneration

*C. elegans* provides a robust system to investigate in vivo axon regeneration at single-neuron resolution [9]. Previously, Rab27 was identified in a large-scale screen as a key inhibitor of regeneration [8]. This work demonstrated that Rab27B/*rab-27* inhibits regeneration in both mouse and *C. elegans* models, and indicated that one site of function for RAB-27 in *C. elegans* is in the injured neurons. However, in *C. elegans*, *rab-27* is highly expressed in the anterior- and posterior-most cells of the intestine as well as the nervous system [10,11]. A potential function of *rab-27* in the intestine was not previously tested.

To study *rab-27*'s function in axon regeneration, we used the same regeneration assay as described in previous work [8]. We used the GABAergic neurons as our model system, lesioning individual axons with a pulsed laser and measuring subsequent regeneration (Fig 1A). As shown previously, loss of *rab-27* resulted in high regeneration, with significant regeneration enhancement occurring as early as 12 hours after axotomy (Fig 1B). *rab-27* mutants produced growth cones earlier and at a higher proportion than in controls, and axons of *rab-27* mutant animals that initiated regeneration grew further and reached the dorsal nerve cord earlier compared to control axons (Fig 1C and 1D).

Next, to determine whether intestinal *rab-27* might function in regeneration, we expressed *rab-27* in either the intestine or the neurons of mutant animals. The intestine is known to signal to the *C. elegans* nervous system to regulate the defecation motor program [10,12,13]. However, signals from the intestine, which must travel through the pseudocoelom to reach the GABAergic neurons, have not previously been implicated in regulation of axon regeneration. We expected that expression in a tissue where it functions would restore normal, lower levels of regeneration. Surprisingly, re-expression of *rab-27* in the intestine of mutants was sufficient to significantly reduce regeneration compared to *rab-27* mutant animals (Figs 1E, 1G, 1I–1K and S1D), indicating that the intestine is a major site of *rab-27* function in inhibiting axon

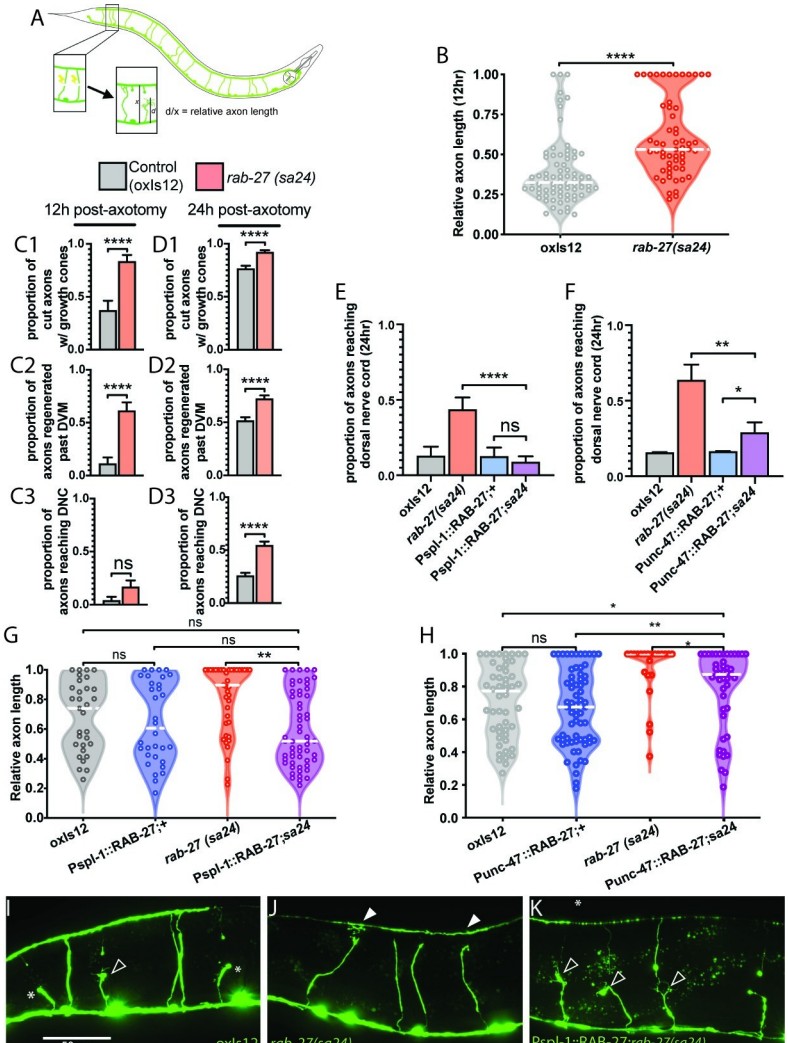

**Fig 1. RAB-27 expression in the intestine inhibits axon regeneration.** (A) Posterior DD/VD commissural axons in the GABAergic nervous system of L4 animals were severed using a pulsed laser, and regeneration was measured after a 24 hour recovery window. (B) Relative axon length in control (*oxIs12*) animals and *rab-27(sa24)* mutants after 12 hours of recovery after axotomy. Axons cut per genotype, L to R: 75, 57. Kolmogorov-Smirnov test was used. ns, not significant, * p < 0.05, *** p < 0.0005. (C). Proportion of cut axons forming growth cones (C1), regeneration past the dorsoventral midline (DVM) (C2), or full regeneration back to the dorsal nerve cord (DNC) (C3) in control (*oxIs12*) and *rab-27(sa24)* mutant animals after 12 hours of recovery post-axotomy. Axons cut per genotype, L to R: 27, 36. Unpaired t-test was used. ns, not significant, **** p < 0.0001. Error bars represent SEM. (D). Proportion of cut axons forming growth cones (D1), regeneration past the dorsoventral midline (DVM) (D2), or full regeneration back to the dorsal nerve cord (DNC) (D3) in control (*oxIs12*) and *rab-27(sa24)* mutant animals after 24 hours of recovery post-axotomy. Axons cut per genotype, L to R: 233, 198. Unpaired t-test was used. ns, not significant, **** p < 0.0001. Error bars represent SEM. (E) Proportion of cut axons showing signs of regeneration in control (*oxIs12*) and *rab-27(sa24)* mutant animals, and animals expressing RAB-27 cDNA under an intestine-specific promoter (Pspl-1) and stabilized with *rab-3* 3' UTR sequence, in both control and *rab-27* mutant backgrounds. Axons were scored after 24 hours of recovery post-axotomy. Axons cut per genotype, L to R: 31, 39, 32, 57. Unpaired t-test was used. ns, not significant, **** p < 0.0001. Error bars represent SEM. (F) Proportion of cut axons showing signs of regeneration in control (*oxIs12*) and *rab-27(sa24)* mutant animals, and animals expressing RAB-27 cDNA under a GABA neuron-specific promoter (Punc-47) and stabilized with *rab-3* 3' UTR sequence, in both control and *rab-27* mutant backgrounds. Axons were scored after 24 hours of recovery post-axotomy. Axons cut per genotype, L to R: 51, 22, 67, 45. Unpaired t-test was used. ns, not significant, * p < 0.05, ** p < 0.005. Error bars represent SEM. (G) Relative axon length in control (*oxIs12*) animals, *rab-27(sa24)* mutants, and animals expressing RAB-27 cDNA under an intestine-specific promoter and stabilized with *rab-3* 3' UTR sequence, in both control and *rab-27* mutant backgrounds. Number of axons cut per genotype, L to R: 31, 32, 39, 57. Kolmogorov-Smirnov test was used. ns, not significant, * p < 0.05, ** p < 0.005. (H) Relative axon length in animals expressing RAB-27 cDNA under a GABA neuron-specific promoter, in

both control (*oxIs12*) and *rab-27* mutant backgrounds. Number of axons cut per genotype, L to R: 51, 67, 22, 45. Kolmogorov-Smirnov test was used. ns, not significant, * p < 0.05, ** p < 0.005, *** p < 0.0005. (I-K). Representative micrographs of regeneration in Day 1 adults 24 hours after axotomy in *oxIs12* control (I), *rab-27* mutant (J), and intestinal *rab-27* rescue (K) animals. Filled arrows indicate fully regenerated axons reaching the dorsal nerve cord, empty arrows indicate partially regenerated axons, and stars indicate non-regenerating axon stumps. All animals express Punc-47::GFP (*oxIs12*).

regeneration. Expression of *rab-27* in the GABA neurons of *rab-27* mutants also reduced regeneration relative to *rab-27* mutant animals, as previously described [8]. Thus, *rab-27* can function in both the intestine and in GABA neurons to inhibit axon regeneration.

Expression of *rab-27* in GABA neurons had a significant effect on regeneration but was not sufficient to fully suppress regeneration to control levels (Figs 1F, S1A and S1C). We had previously found that expressing *rab-27* in GABA neurons restores regeneration to control levels [8]. Our current strategy to express *rab-27* only in GABA neurons used an expression construct that contained the *rab-3* 3'UTR, while our previous efforts used the unc-54 3'UTR. The unc-54 UTR sequence can itself drive expression in the posterior gut because it contains regulatory and coding sequence for the intestinal gene *aex-5* [14]. We hypothesized that a requirement for intestinal expression accounts for the different effects of the UTR. Intestine-specific *rab-27* rescue constructs containing the *rab-3* 3'UTR rescued axon regeneration identically to those containing the unc-54 3'UTR (S1B Fig). Use of the *rab-3* 3' UTR in the intestine-specific RAB-27 rescue construct also produced a much stronger rescue of *rab-27* mutants' aex phenotype, with nearly full restoration of the pBoc/expulsion ratio, compared to only a partial rescue by constructs containing the unc-54 3' UTR (S2 Fig). Thus, *rab-27* can act in either neurons or the intestine to suppress regeneration, but intestinal expression is necessary for complete function. Overall, these tissue-specific experiments raise the question of whether similar or different cellular mechanisms mediate *rab-27*'s regeneration function in these two tissues.

## RAB-27's synaptic vesicle tethering cofactors do not inhibit regeneration

In neurons, *rab-27* is thought to function similar to the well-studied Rab family member *rab-3*. Phylogenetic analysis of the *C. elegans* Rab family shows that *rab-27* and *rab-3* are each other's closest paralog [15]. RAB-3 and RAB-27 are both enriched in the nerve ring of *C. elegans* [16], suggesting synaptic localization, and both Rabs colocalize at synapses in mammalian neurons [17]. Consistent with these studies, we found that tagged *rab-3* and *rab-27* colocalize at synapses in *C. elegans* GABA neurons (Fig 2A). *rab-3* regulates synaptic vesicle tethering and synaptic transmission [16], and *rab-27* is thought to play an auxiliary role in this process [16,17]. Further, both *rab-27* and *rab-3* are regulated by a common GEF MADD/*aex-3*, and *aex-3* is required for normal synaptic transmission [16]. However, despite these similarities, other data suggest that *rab-27* and *rab-3* also have different functions. In *C. elegans*, the Rab effector protein Rabphilin/*rbf-1* genetically interacts with *rab-27* but not *rab-3* [16,18–20]. Further, *rab-27* and *rbf-1*, but not *rab-3*, are required for tethering and secretion of dense core vesicles in neurons [21–24]. Finally, *rab-27*, unlike *rab-3* or Rabphilin/*rbf-1*, is expressed in both neurons and intestine [11,19]. Consistent with this, *rab-27* mutants but not *rab-3* or Rabphilin/*rbf-1* mutants have a constipated phenotype due to a defect in dense core vesicle release from the intestine and resulting disruption of the defecation motor program (DMP) [25]. These data raise the question of what the relationship is between *rab-27* and *rab-3* in axon regeneration.

We used genetic analysis to determine the relationship between *rab-27*, *rab-3* and the RAB-27 effector Rabphilin/*rbf-1* in axon regeneration. Loss of *rab-3* did not affect axon regeneration (Fig 2B), suggesting that unlike for synaptic vesicle release, where *rab-3* predominates [16], *rab-27* rather than *rab-3* is the major factor in axon regeneration. Loss of Rabphilin/*rbf-1* also

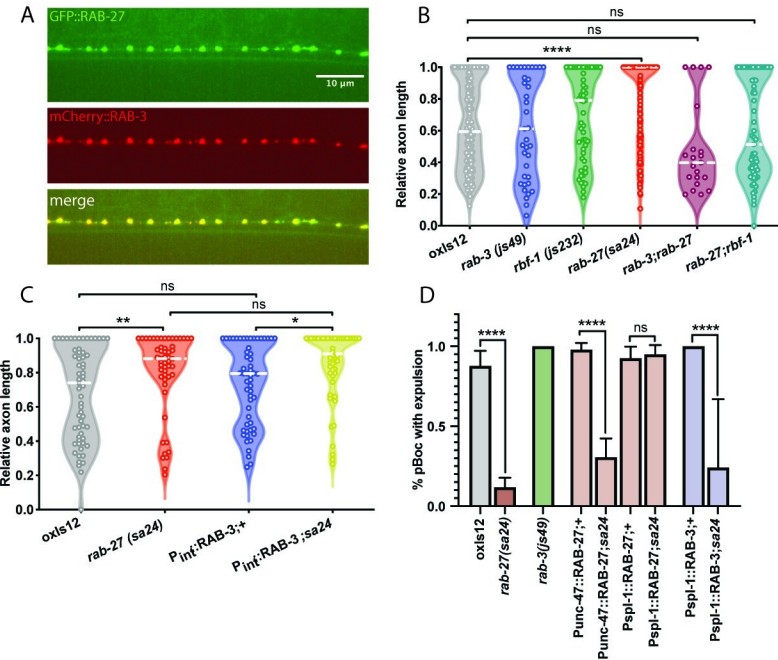

**Fig 2. RAB-27's synaptic vesicle tethering cofactors do not inhibit regeneration.** (A) Colocalization of transgenic GFP::RAB-27 and mCherry::RAB-3 at synapses of DD/VD neurons. (B) Relative axon length in control (*oxIs12*) animals, *rab-3*(*js49*), rbr-1(*js232*), *rab-27*(*sa24*), *rab-3*(*js49*); *rab-7*(*sa24*) mutants. Axons cut per genotype, L to R: 183, 37, 55, 196, 21, 69. Kolmogorov-Smirnov test was used. ns, not significant, * $p < 0.05$, *** $p < 0.0005$. (C) Relative axon length in control animals, *rab-27*(*sa24*) mutants, and animals expressing RAB-3 cDNA under an intestine-specific promoter, in control and *rab-27* mutant backgrounds. Number of axons cut per genotype, L to R: 61, 55, 53, 50. Kolmogorov-Smirnov test was used. ns, not significant, * $p < 0.05$, ** $p < 0.005$. (D) Mutants in the *aex* pathway display a defect in the defecation motor program, visualized by a loss of waste expulsion (Exp) following posterior body contraction (pBoc). D1 adult animals were randomly selected and observed for 5 DMP cycles, and the ratio of Exp/pBoc was plotted. Intestinal (P*spl-1*) but not GABA neuron-specific (P*unc-47*) expression of *rab-27* cDNA was sufficient to rescue DMP in *rab-27* mutant worms. Expression of *rab-3* cDNA in the intestine of *rab-27* mutant animals did not rescue DMP defects. pBoc cycles observed, L to R: 49, 119, 30, 49, 62, 54, 56, 40, 58. ns, not significant, * $p < 0.05$, ** $p < 0.05$, *** $p < 0.0005$, **** $p < 0.0001$, Fisher's Exact Test. Error bars represent SEM.

did not affect regeneration. However, double mutants for either *rab-27*;*rab-3* or *rab-27*;*rbf-1* suppressed the high regeneration phenotype of *rab-27* single mutants (Fig 2B). We conclude that a neuronal function mediated by *rab-3* and Rabphilin/*rbf-1* is required for enhanced regeneration in *rab-27* mutants, though this neuronal function is dispensable for normal regeneration.

A major site of *rab-27* function in axon regeneration is the intestine (Fig 1G), where *rab-3* is not expressed [26]. Given the close evolutionary and functional relationship between *rab-27* and *rab-3*, it is possible that *rab-3* could function in the intestine to inhibit axon regeneration, but is simply not expressed there. To test this idea, we ectopically expressed RAB-3 in the intestine of *rab-27* mutants to see whether RAB-3 could compensate for loss of *rab-27*. Intestinal expression of RAB-3 in *rab-27* mutants was not sufficient to rescue high regeneration (Fig 2C). Intestinal RAB-3 also failed to rescue DMP defects in *rab-27* mutants (Fig 2D). Thus, for the two distinct phenotypes of axon regeneration and DMP, *rab-27* mutants expressing intestinal RAB-3 were indistinguishable from non-transgenic *rab-27* mutants. By contrast, intestinal re-expression of RAB-27 cDNA in *rab-27* mutants showed a significant rescue of DMP defects (Figs 2D, S2A and S2B), in addition to restoring normal levels of axon regeneration (Fig 1G). Together, these results indicate that despite their similarity and shared function in synaptic

vesicle tethering, RAB-27 and RAB-3 are functionally distinct, and raise the question of what mechanisms act with RAB-27 to mediate its intestinal function in axon regeneration.

## Intestinal components of a secretory vesicle signaling pathway inhibit regeneration

In the intestine, *rab-27* acts to facilitate the tethering and fusion of dense core vesicles during the defecation motor program (DMP) [19]. At the expulsion (*Exp*) step of the DMP, a neuropeptide ligand packaged into DCVs is secreted from the intestine. This peptide signal is sensed by receptors on the GABAergic neurons AVL and DVB, which in drive contractions of the enteric muscles and eventually waste expulsion [10,13,25]. Packaging and fusion of these intestinal DCVs involves *rab-27*, together with the pro-protein convertase KPC3/*aex-5*, the t-SNARE protein SNAP25/*aex-4*, the Munc13-like SNARE regulator *aex-1*, the Rab GEF recruitment factor NPDC1/*cab-1*, and the Rab GEF MADD/*aex-3* [27,28]. The neuronal receptor that responds to neuropeptide release from the intestine is the GPCR *aex-2*. Loss of function in any of these genes disrupts the DMP and results in a constipation phenotype [10,13,25], while intestinal re-expression is sufficient to significantly restore normal DMP function (S3 Fig).

We hypothesized that this same DCV secretion mechanism may account for *rab-27*'s function in axon regeneration. Consistent with this hypothesis, we found that NPDC1/*cab-1*, KPC3/*aex-5*, and SNAP25/*aex-4* all inhibit axon regeneration (Figs 3A and 4A). As in *rab-27* mutants, intestine-specific re-expression of each of these genes in their respective mutant backgrounds was sufficient to rescue normal regeneration (Figs 3B and 4B), supporting an intestinal origin of regeneration inhibition. The regeneration phenotypes of *cab-1* and *aex-4* mutants is similar to *rab-27* mutants, and both double mutants of *rab-27*;*cab-1* and *rab-27*; *aex-4* do not show further enhancement of axon regeneration (Fig 4C), consistent with a cellular role for NPDC-1/*cab-1* upstream and for SNAP25/AEX-4 downstream of RAB-27. Mutants for KPC3/*aex-5* by contrast showed a less dramatic, though still significant, improvement in axon regeneration, while loss of the Rab GEF MADD/*aex-3* or SNARE regulator Munc13-b/ *aex-1* did not significantly affect regeneration (Fig 3A). Together, these results indicate that neuropeptide processing and vesicle secretion from the intestine are important for axon regeneration inhibition.

Over 250 distinct neuropeptides have been identified in *C. elegans* [29], of which approximately fifty are believed to be expressed in the intestine [11]. NLP-40 has previously been identified as a neuropeptide that is specifically expressed in the intestine [13,30], and signals to the nervous system to regulate the DMP [13]. NLP-40 is the secreted signal linking the intestine to the GABAergic neurons AVL and DVB in the DMP [10,13], is essential for normal waste expulsion, and its secretion is dependent on SNAP25/AEX-4 [13]. Loss of *nlp-40* lead to a mild but significant increase in axon regeneration success (Fig 5A). High regeneration in *nlp-40* mutants was similar to the regeneration phenotype of KPC3/*aex-5* (Fig 3A and 3B), and was rescued by intestine-specific NLP-40 cDNA re-expression (Fig 5B), suggesting that NLP-40 may work in the intestine to regulate axon regeneration. The *nlp-40* regeneration phenotype was similar to that of KPC3/*aex-5* mutants (Fig 3A and 3B), suggesting that the two signals may work together to regulate regeneration, as they do to regulate waste expulsion. Additionally, the relatively mild improvement in regeneration success seen in *nlp-40* and KPC3/*aex-5* mutants suggests that NLP-40 processing and secretion may be only one of several mechanisms by which the intestine regulates regeneration. A small candidate screen of other intestinally-expressed neuropeptide-like proteins (NLPs) that are expressed in the intestine and are processing targets of KPC3/AEX-5 (29) did not identify any additional inhibitors of regeneration (Fig 5A).

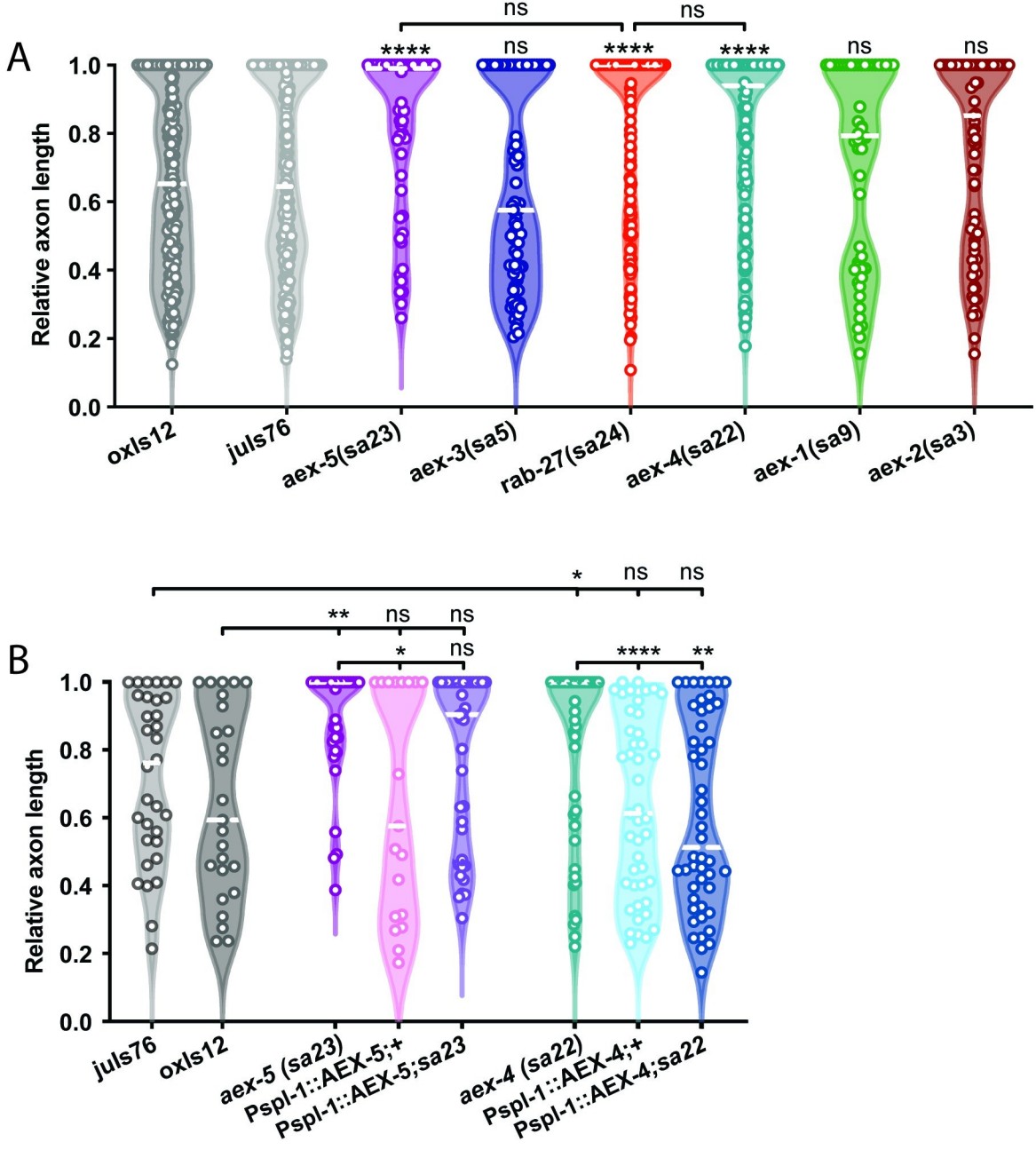

**Fig 3. AEX-4 and AEX-5 inhibit axon regeneration.** (A) Relative axon length in control animals expressing GABAergic neuron-specific GFP (*oxIs12* & *juIs76*), and *aex-1*(sa9), *aex-2*(*sa3*), *aex-3*(*sa5*), *aex-4*(*sa22*), *aex-5*(*sa23*) and *rab-27*(*sa24*) mutants. *aex-1*, *aex-5*, and *rab-27* are compared against *oxIs12*, while *aex-2*, *aex-3*, while *aex-4* are compared against *juIs76*. Axons cut per genotype, L to R: 238, 199, 37, 83, 148, 69, 50, 66. Kolmogorov-Smirnov test was used. ns, not significant, * p < 0.05, ** p < 0.005 **** p < 0.0001. (B) Relative axon length in control animals expressing GABAergic neuron-specific GFP (*oxIs12* & *juIs76*), *aex-5*(*sa23*) and *aex-4*(*sa22*) mutant animals, and animals expressing AEX-5 and AEX-4 cDNA under an intestine-specific promoter (*Pspl-1*) and stabilized with *rab-3* 3' UTR sequence, in both control and respective mutant backgrounds. Axons cut per genotype, L to R: 32, 25, 19, 45, 46, 48, 33, 38. Kolmogorov-Smirnov test was used. ns, not significant, * p < 0.05, ** p < 0.005 **** p < 0.0001.

*C. elegans* has between 125 to 150 G-protein coupled neuropeptide receptor homologs [31,32], of which approximately 20 are expressed in the DD/VD GABAergic motor neurons [33]. Of these, AEX-2 is a known GCPR for NLP-40 involved in AVL/DVB activation during

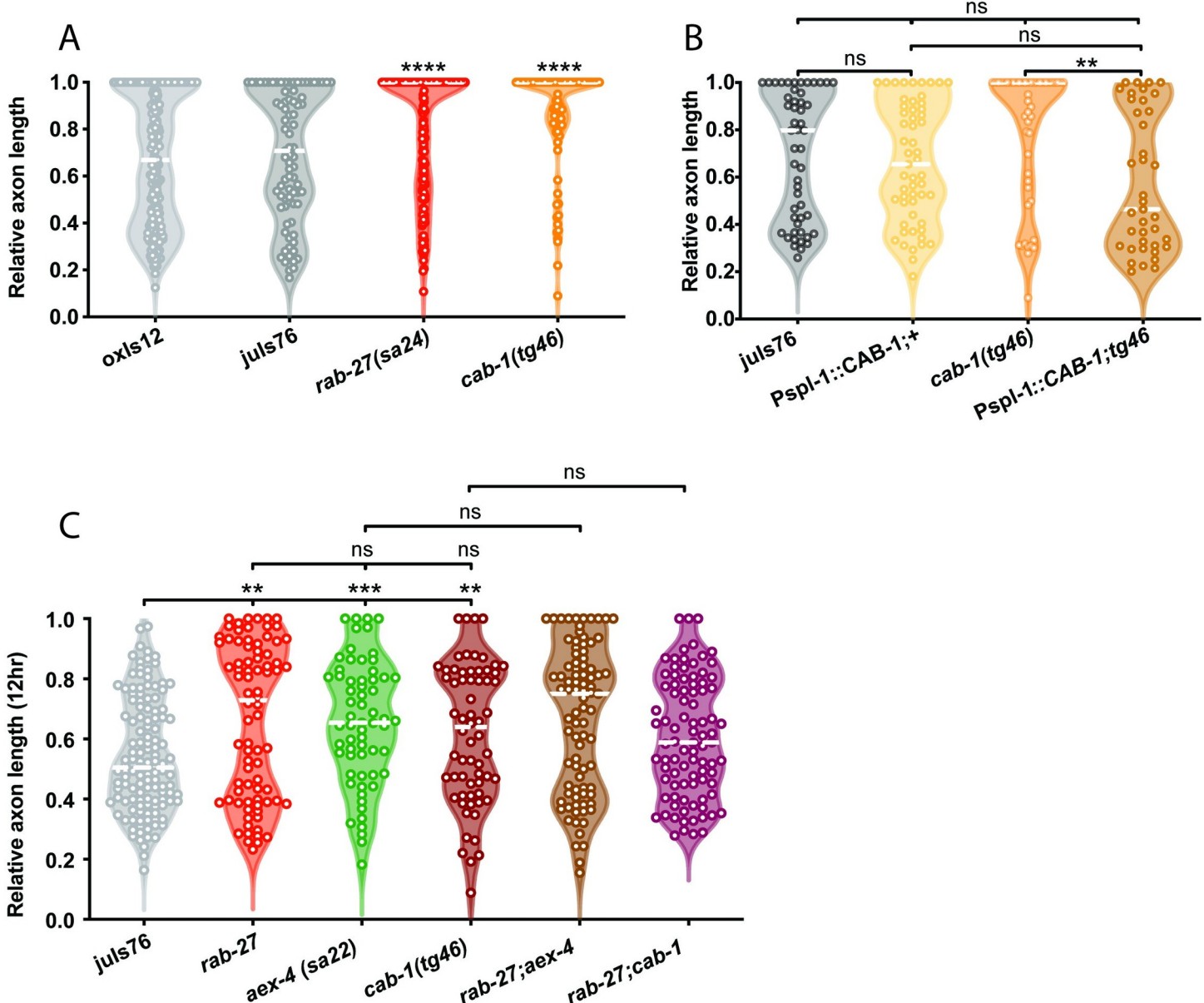

**Fig 4. CAB-1 inhibits axon regeneration.** (A) Relative axon length in control animals expressing GABAergic neuron-specific GFP (*oxIs12* & *juIs76*), and *rab-27*(*sa24*) and *cab-1*(*tg46*) mutants. *rab-27* is compared against *oxIs12*, while *cab-1* is compared against *juIs76*. Axons cut per genotype, L to R: 200, 81, 164, 91. Kolmogorov-Smirnov test was used. ns, not significant, **** p < 0.0001. (B) Relative axon length in control (*juIs76*) animals, *cab-1*(*tg46*) mutants, and animals expressing CAB-1 cDNA under an intestine-specific promoter and stabilized with *rab-3* 3' UTR sequence, in both control and *cab-1* mutant backgrounds. Number of axons cut per genotype, L to R: 52, 55, 50, 39. Kolmogorov-Smirnov test was used. ns, not significant, ** p < 0.005. (C) Relative axon length in control animals expressing GABAergic neuron-specific GFP (*juIs76*), *rab-27*(*sa24*), *aex-4*(*sa22*), *cab-1*(*tg46*) mutants and *rab-27*(*sa24*); *aex-4*(*sa22*) and *rab-27*(*sa24*);*cab-1*(*tg46*) double mutants. Axons cut L to R: 127, 80, 63, 64, 87, 90. Kolmogorov-Smirnov test was used. ns, not significant, ** p < 0.005. Regeneration was scored after 12 hours of recovery to more easily visualize enhanced regeneration in the *rab-27* and *rab-27*;*cab-1* double mutants, which show nearly full regeneration after the usual 24 hour recovery window.

defecation [13]. We find that loss of *aex-2* does not result in significant enhancement of regeneration (Fig 3A). This suggests that NLP-40 and the RAB-27-dependent pathway work partially or entirely through a separate neuronally-expressed GPCR, which is further supported by AEX-2's expression, which in GABAergic neurons appears largely limited to AVL and DVB (Mahoney et al. 2008), and is not strongly expressed in the DD or VD neurons [33]. The

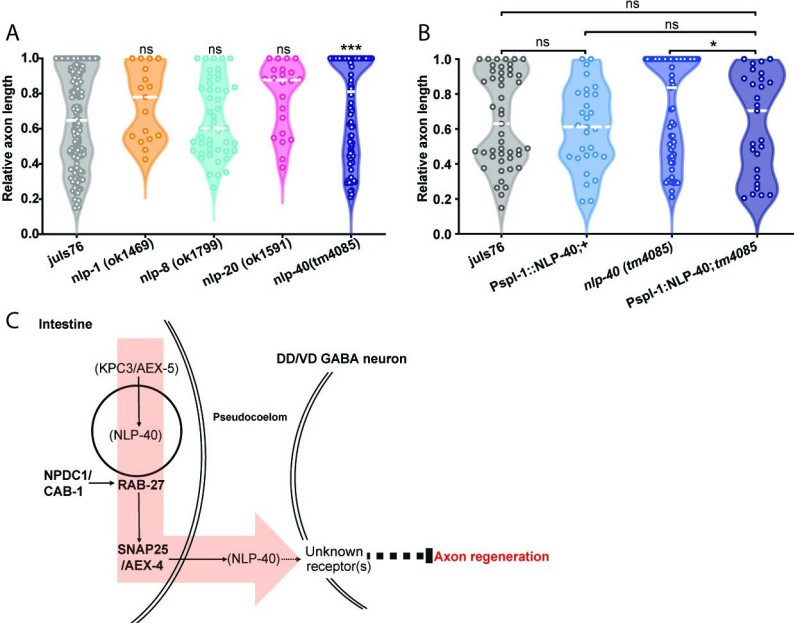

**Fig 5. The neuropeptide NLP-40 inhibits axon regeneration.** (A) Relative axon length in control animals expressing GABAergic neuron-specific GFP (*juIs76*), and mutants of several intestinally-expressed neuropeptides: *nlp-1(ok1469)*, *nlp-8(ok1799)*, *nlp-20(ok1591)* and *nlp-40(tm4085)*. Axons cut per genotype, L to R: 117, 17, 47, 22, 67. Kolmogorov-Smirnov test was used. ns, not significant, **** p < 0.0001. (B) Relative axon length in control (*juIs76*) animals, *nlp-40 (tm4085)* mutants, and animals expressing sequence for mature NLP-40 peptide under an intestine-specific promoter and stabilized with *rab-3* 3' UTR sequence, in both control and *nlp-40* mutant backgrounds. Number of axons cut per genotype, L to R: 47, 65, 29, 27. Kolmogorov-Smirnov test was used. ns, not significant, ** p < 0.005. C) Schematic of axon regeneration inhibition from the intestine. NPDC1/CAB-1, RAB-27 and the SNAP25 ortholog AEX-4 form a pathway regulating the tethering and fusion of dense core vesicles at the basal membrane of the intestinal cells. Cargo in these vesicles are secreted into the pseudocoelom, where they signal to GABAergic neurons to inhibit regeneration, through a currently unknown receptor or series of receptors. One secreted inhibitory cargo is the neuropeptide NLP-40, which is processed by the proprotein convertase KPC3/AEX-5, and is secreted into the pseudocoelom through a SNAP25/AEX-4-dependent mechanism. Disruption in any of these genes leads to enhanced regeneration of the DD/VD GABAergic neurons.

identity of additional peptide signals, as well as the neuronally-expressed receptor or receptors remain unknown. Further work is required to identify these components of the intestine-neuron signaling axis that inhibits axon regeneration.

## Multiple Rab GTPases affect axon regeneration

*rab-27* was initially identified as a candidate regeneration inhibitor in a functional genome-wide screen for regeneration inhibitors done in mammalian cortical neurons in vitro that identified 19 Rab GTPases as potential regeneration inhibitors [8]. *C. elegans has* a drastically reduced cohort of functional Rabs compared to mammals [15], attributable in large part to decreases in redundancy. Compared to the results seen in mammalian cell culture, a few Rabs in *C. elegans* affect regeneration (Fig 6A). In addition to *rab-27* and the previously identified *rab-6.2* [34], loss of *rab-18* significantly decreases regeneration success, while loss of *glo-1* leads to a modest increase in regeneration. Unlike other high-regenerating Rab mutants, *glo-1* mutants specifically show an increase in full regeneration after 24 hours of recovery, though not an increase in the likelihood of regeneration initiation during that period (Fig 6B and 6C). GLO-1 is expressed specifically in the intestine, where it localizes to and is required for the biogenesis of the lysosome-like gut granules [35]. Along with *rab-27*, the effect of glo-1 on

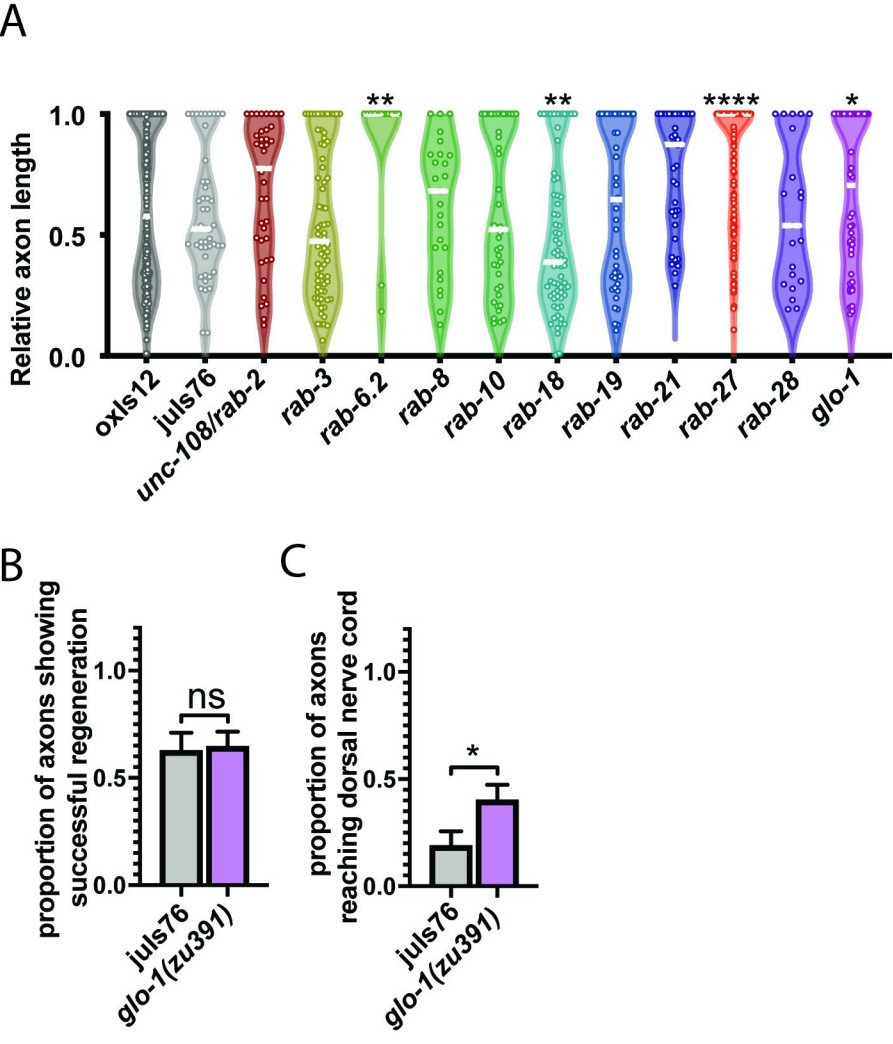

**Fig 6. Multiple Rab GTPases affect axon regeneration.** (A) Relative axon length in control animals expressing GABAergic neuron-specific GFP (*oxIs12* & *juIs76*), and unc-108/rab-2(n501), *rab-3(js49)*, *rab-6.2(ok2254)*, *rab-8 (tm2526)*, *rab-10(q373)*, *rab-18(ok2020)*, *rab-19(ok1845)*, *rab-21(gk500186)*, *rab-27(sa24)*, *rab-28(gk1040)*, and *glo-1 (zu391)*. *unc-108/rab-2*, *rab-3*, *rab-8*, *rab-10*, *rab-18*, *rab-19*, *rab-21*, *rab-27* and *rab-28* are compared against *oxIs12*, while *rab-6.2* and *glo-1* are compared against *juIs76*. Axons cut per genotype, L to R: 396, 46, 39, 72, 13, 25, 41, 69, 43, 38, 123, 21, 45, 64. Kolmogorov-Smirnov test was used. ns, not significant, * p < 0.05, ** p < 0.005 **** p < 0.0001. (B) Proportion of cut axons showing signs of regeneration in control (*juIs76*) and *glo-1(zu391)* mutant animals. Axons cut per genotype, L to R: 32, 45. Unpaired t-test was used. ns, not significant. Error bars represent SEM. (C) Proportion of cut axons showing full regeneration back to the dorsal nerve cord in control (*juIs76*) and *glo-1(zu391)* mutant animals. Axons cut per genotype, L to R: 32, 45. Unpaired t-test was used. ns, not significant. Error bars represent SEM.

regeneration suggests that the intestine may play a previously unknown but important role in regulation of axon regeneration.

## Discussion

Axon regeneration is tightly regulated by pathways from within the injured neuron as well as by interactions with the local environment, but the existence of long-range regulatory signals has remained unclear. Here we show that in *C. elegans*, RAB-27 acts in the intestine to inhibit regeneration of severed axons of the DD/VD GABAergic motor neurons. This inhibition

occurs independently of *rab-27*'s known role in neurons, where it regulates synaptic vesicle fusion and also functions in axon regeneration [8,16].

We find that multiple factors involved in dense core vesicle (DCV) packaging and secretion from the intestine inhibit regeneration along with *rab-27*. CAB-1 and SNAP25/AEX-4, which functions in DCV trafficking and fusion (E. Jorgensen, pers. comm., 14,39), as well as KPC3/AEX-5, which functions in neuropeptide processing [36], and the neuropeptide NLP-40, all significantly inhibit regeneration. Intestine-specific re-expression of each of these genes is sufficient to restore normal levels of axon regeneration, pointing to an intestinal origin of regeneration inhibition. These data suggest a model in which axon regeneration is regulated through secretion of inhibitory ligands from the intestine. This mechanism of regeneration is shared in part with the vesicle secretion and neuropeptide release pathway used for regulation of the defecation motor program [10,13] (Fig 5C). Because the DMP pathway is constitutively active in adult animals, and NLP-40 in particular is constitutively secreted from the intestine [13], the pathway we describe of intestinal regeneration inhibition is likely not specifically triggered by axon injury, but is more likely incidental to the primary function of this pathway.

The strong improvement of regeneration seen in *rab-27*, *aex-4*, and *cab-1* mutants, which is not seen in all components of the intestinal DMP pathway, raises the possibility that these genes may be involved in the secretion of multiple inhibitory signals from the intestine, of which NLP-40 is only one candidate. Conversely, the comparatively weak regeneration phenotypes seen in *nlp-40* and KPC3/*aex-5* mutants suggest that NLP-40 processing and secretion may only represent a part of the inhibitory cargo secreted by this pathway. Identification of additional inhibitory ligands from the intestine will rely on a more complete understanding of the diversity of intestinal vesicles whose secretion is regulated by CAB-1, RAB-27 and AEX-4, as well as understanding the nature of identified inhibitory ligands. It seems unlikely that the mature *C. elegans* intestine expresses and secretes a signal dedicated to post-developmental axon growth inhibition; a more attractive hypothesis is that regeneration inhibition is an incidental consequence of an unrelated homeostatic signal, or possibly a relic effect of secreted signals involved in intestinal development, which is known to rely on signals also involved in axon guidance [37].

Surprisingly we do not find a regeneration phenotype for Munc-13b/*aex-1* (Fig 3A). Munc13 proteins are involved in SNARE-mediated vesicle docking and fusion [38,39], and Munc13-b/*aex-1* loss disrupts intestinal DCV fusion and leads to significant DMP defects [39]. These data suggest that the intestinal DCV population that mediates regeneration is distinct from DCVs that mediate the DMP. Presumably the "regeneration DCVs" rely on a different factor than the "DMP DCVs" to mediate SNARE-directed fusion. However, we did not detect a role in regeneration for CAPS/*unc-31* (S4A Fig), another factor that mediates SNARE-directed membrane fusion (39). One possibility is that Munc-13b/AEX-1 may function redundantly with other vesicle docking regulators to mediate DCV fusion for axon regeneration. Coregulation of DCV fusion between AEX-1 and other factors could also conceal involvement of this important DCV regulator in regeneration. Alternatively, it is possible that we may have failed to detect a subtle regeneration effect in *aex-1* mutants.

HID-1 is an important component of DCV-dependent neurosecretion, and like RAB-27 is expressed both in the nervous system and the intestine, where it regulates localization of the effector Rabphilin/*rbf-1* [19]. Loss of *hid-1* leads to several defects similar to *rab-27* mutants, including constipation, reduced locomotion and egg-laying defects, and has been shown to act in a parallel pathway to RAB-3 and RAB-27 [19]. Despite these expression and phenotypic similarities, analysis of *hid-1* mutants showed not significant regeneration phenotypes (S4B Fig), suggesting that HID-1 is not required in the transmission of an inhibitory signal.

In the nervous system, RAB-27 regulates synaptic vesicle tethering in coordination with the closely related RAB-3, upstream of the effector Rabphilin/RBF-1 [16,19]. While neuronal RAB-27 inhibits regeneration (Fig 1H), loss of *rab-3* or Rabphilin/*rbf-1* does not affect regeneration (Fig 2B). These data suggest that neuronal RAB-27 inhibits axon regeneration independent of its role in synaptic vesicle tethering. As it does in diverse tissues across species, RAB-27 also regulates the tethering and fusion of non-synaptic vesicles in *C. elegans* neurons [23], and as with the intestine, neuronal RAB-27 may regulate the secretion of an unknown ligand or ligands through dense core vesicles to inhibit regeneration. However, important differences underlie the potential inhibitory mechanisms of neuronal and intestinal RAB-27-dependent regeneration inhibition. As we have shown, intestinal RAB-27 mediates regeneration inhibition in part through a specific pathway that regulates homeostatic intestine-to-neuron communication, and relies on several components, such as SNAP25/*aex-4* and *nlp-40* that are exclusively expressed in the *C. elegans* intestine [11]. Several possibilities could explain neuronal RAB-27's incomplete rescue of high regeneration compared to intestinal RAB-27: the two tissue-specific RAB-27-dependent pathways may be regulating the release of different inhibitory ligands or ligand cohorts, with the intestine secreting the more potent inhibitor(s). Alternatively, intestinal and neuronal RAB-27 could regulate the release of the same inhibitory signals, but through distinct secretory pathways of different effectiveness.

While loss of *rab-3* or Rabphilin/*rbf-1* alone does not affect regeneration, loss of either in a *rab-27* mutant background completely suppresses the *rab-27* mutant high regeneration phenotype (Fig 2B). However, these double mutants, which show severe defects in synaptic transmission [16], do not show any defects in regeneration beyond the suppression of the *rab-27* mutant phenotype (Fig 2B). These data suggest that robust synaptic vesicle fusion is required only for enhanced regeneration. Significant loss of vesicle fusion below a certain threshold may restrict high regeneration by restricting the available pool of membrane required for enhanced outgrowth [40]. Alternatively, loss of synaptic vesicle tethering and fusion could disrupt specific pro-regeneration pathways that are normally inhibited during regeneration, but that are released following loss of inhibitory upstream regulatory signals such as RAB-27. Thus, neuronal RAB-27 appears to have dual roles in the regulation of axon regeneration: a pro-high regenerative role mediated through synaptic vesicle fusion and co-regulated by RAB-3 and Rabphilin/RBF-1, and an inhibitory role mediated by the secretion of an anti-regeneration signal from DCV fusion.

Rab GTPases are emerging as key regulators of axon regeneration in vitro and in vivo. *C. elegans* provides an excellent system to probe the "rabome" for novel pathways affecting axon regeneration. In *C. elegans*, *rab-6.2* was previously shown to affect regeneration [34], as was *rab-27* function in neurons [8]. This work probed the function of RAB-27 outside the nervous system, revealing an unexpected role for DCV fusion in the intestine in regulation of axon regeneration. Rabs mediate many complex biological processes, such as Parkinson's disease pathogenesis [41] and cancer metastasis through regulation of exosome secretion [42]. This study adds to our understanding of Rab function by identifying a novel role for RAB-27 in mediating a long-range signal that inhibits the ability of neurons to regenerate after injury.

## Materials and methods

### C. elegans strains

Strains were maintained at 20°C, as described in Brenner, 1974 [43] on NGM plates seeded with OP50. Some strains were provided by the CGC, which is funded by the NIH Office of Research Infrastructure Programs (P40 OD010440). The following strains were purchased from the CGC:

NM791[*rab-3(js49)*], RT2[rab-10(e1747)], RB1638[rab-18(ok2020], RB1537[rab-19 (ok1845], JT24[*rab-27(sa24)*], JT699[*rab-27*(sa699)], JJ1271[glo-1(zu391)], VC2505[rab-28 (gk1040)], MT1093[unc-108(n501)], JT23[*aex-5(sa23)*], JT3[*aex-2(sa3)*], JT5[*aex-3(sa5)*], JT9 [*aex-1*(sa9)], KY46[*cab-1(tg46)*], NM1278[*rbf-1(js232)*], NM2777 [aex-6(*sa24*);*rab-3(js49)*]. The following strains were purchased from the NBRP: rab-8(tm2526), nlp-40(tm4085). A complete list of generated strains is available in S1 Table.

## Constructs and cloning

Transgenic constructs were generated with Gateway recombination (Invitrogen). Fluorescent-tagged RAB-27 was generated through fusion PCR [44]. Constructs were microinjected to generate transgenic animals as described in Mello et al., 1991 [45]. Constructs were injected at a concentration of 7.5 ng/μL unless otherwise mentioned. Adult P0 worms were singled onto plates following injection, incubated at 20˚C for three to four days. F1 animals expressing the fluorescent coinjection marker (Pmyo-2::mCherry) were subsequently singled, and stably transmitting lines were established.

## Laser axotomy

Laser axotomy was performed as previously described in Byrne et al., 2014 [46]. L4 animals were immobilized using 0.05 μm polystyrene beads (Polybead Microspheres, Polysciences Cat #08691–10) or in 0.2mM Levamisole (Sigma) on a pad of 3% agarose dissolved in M9 buffer on a glass slide. Worms were visualized using a Nikon Eclipse 80i microscope with a 100x Plan Apo VC lens (1.4 NA). Fluorescently-labeled D-type motor neuron commissures were targeted at the dorsoventral midline using a 435 nm Micropoint laser with 10 pulses at 20 Hz. In all cases no more than four of the seven posterior commisures were cut per animal to minimize possible adverse locomotion or behavioral effects. Animals were recovered to NGM plates seeded with OP50 and allowed to recover.

## Fluorescence microscopy and regeneration scoring

Animals with cut axons were immobilized using 0.25–2.5 mM levamisole (Santa Cruz, sc-205730) and mounted on a pad of 3% agarose in M9 on glass slides. All animals were imaged to visualize regeneration using an Olympus DSU mounted on an Olympus BX61 microscope, with a Hamamatsu ORCA-Flash4.0 LT camera, and Xcite XLED1 light source with BDX, GYX and RLX LED modules. Images were acquired as 0.6 um z-stacks using consistent exposure time, camera sensitivity and light intensity. Images were exported as tiff files and analyzed in ImageJ. Axon regeneration was scored at 24 hours post-axotomy apart from the datasets in Figs 1B and 4B, which were analyzed 12 hours post-axotomy using the same analysis and scoring strategy. Analysis of regeneration at 12 hours was done in double mutants where the regeneration of each single mutant after 24 hours was >95%, with >50% of axons fully regenerated to the DNC, in order to more easily visualize differences in regeneration success. Cut axons were scored based on regeneration status and length, and each individual axon was given a designation showing presence of a growth cone indicative of regeneration initiation (Y,N), its general elongation status (no regeneration, GC below midline, GC at midline, GC above midline, full regeneration to DNC), and the measured axon length (absolute axon growth relative to the distance between dorsal and ventral nerve cords).

For imaging of GFP::RAB-27; mCherry::RAB-3 in intact axons (Fig 2A), worms were immobilized as described above, and imaged using the vt-iSIM system mounted on a Leica DMi8 inverted platform, with a Hammamatsu ORCA-Flash 4.0 camera. Images were acquired as 0.6 um z-stacks using consistent exposure time, camera sensitivity and light intensity.

### Analysis of defecation motor program defects

Day 1 adult worms were scored for defects in the defecation motor program by examining the proportion of waste expulsion (Exp) events to posterior body wall muscle contractions (pBoc), or by visually scoring the severity of waste accumulation in the intestinal lumen. Individual worms were sorted onto blinded plates, and watched for a series of 5 to 10 DMP cycles. Time between pBoc contractions was measured as well as the presence or absence of aBoc contraction and waste expulsion. Inter-cycle waste expulsion (Exp >10 seconds after pBoc contraction, and occurring without immediate prior aBoc contraction), which often occurs in severely constipated *aex* pathway mutants, was not scored as an Exp event. Mean Exp/pBoc ratio was calculated, and statistical comparisons were made using Fisher's Exact Test. For visual scoring of waste accumulation, 20 worms were placed onto blinded plates: 10 transgenic worms expressing a rescue construct (identifiable through pharyngeal mCherry expression), and 10 non-transgenic siblings. Worms were scored as "normal", "constipated" or "slightly constipated" under white light, and genotypes were assessed after sorting.

### Fecundity

L4 worms of each genotype were singled onto NGM plates seeded with 100μL OP50 for 48 hours at 20˚C. Adult worms were removed, and surviving progeny (L1 or older animals) were counted after an additional 24 hours. Unhatched eggs were not counted.

### Graphing and statistical analysis

Data was plotted using Prism 9.2.0 for MacOS. P values for relative regeneration outcomes (violin plots in Figs 1B,1G and 1H, S1A and S1B, 2B and 2C, 3, S4, 4A and 4B, 5A and 5B and 6A) were calculated using the Kolmogorov-Smirnov test, with all scoring done blinded to genotype. In violin plots, the median is represented by a dashed white line. In cases where the median lies at 1.0 (where >50% of axons have fully regenerated), the bar may be partially obscured by accumulated individual measurements. Differences in qualitative regeneration analysis (Figs 1C–1F, S1C and S1D, 6B and 6B), aex phenotype rescue (Figs 2D and S2), and fecundity (S5 Fig) were calculated using Fisher's exact test.

### Supporting information

**S1 Fig. Use of *unc-54* 3' UTR sequence in constructs containing RAB-27 cDNA inhibits regeneration.** (A-B) Relative axon length in animals expressing RAB-27 cDNA under a GABA neuron-specific (A) or intestine-specific (B) promoter and with *unc-54* 3' UTR sequence, in both control (*oxIs12*) and *rab-27* mutant backgrounds. Number of axons cut per genotype, L to R: 51, 67, 22, 45. Kolmogorov-Smirnov test was used. ns, not significant, * $p < 0.05$, ** $p < 0.005$, *** $p < 0.0005$. (C) Proportion of cut axons showing signs of successful regeneration initiation (C1) or regeneration past the dorsoventral midline (C2) in control (*oxIs12*) and *rab-27(sa24)* mutant animals, and animals expressing *rab-27* cDNA under a GABA neuron-specific promoter (Punc-47) and the *rab-3* 3' UTR sequence, in both control and *rab-27* mutant backgrounds. Axons were scored after 24 hours of recovery post-axotomy. Axons cut per genotype, L to R: 51, 22, 67, 45. Unpaired t-test was used. ns, not significant. Error bars represent SEM. (D) Proportion of cut axons showing signs of successful regeneration initiation (D1) or regeneration past the dorsoventral midline (D2) in control (*oxIs12*) and *rab-27(sa24)* mutant animals, and animals expressing *rab-27* cDNA under an intestine-specific promoter (Pspl-1) and the *rab-3* 3' UTR sequence, in both control and *rab-27* mutant backgrounds. Axons cut per genotype, L to R:

31, 39, 32, 57. Unpaired t-test was used. ns, not significant, ** p < 0.005. Error bars represent SEM.
(TIF)

**S2 Fig. Rescue of the defecation motor program by intestinal *rab-27* expression.** (A) pBoc Re-expression of RAB-27 rescue constructs in the GABAergic neurons were did not rescue DMP defects. Intestinally-expressed RAB-27 cDNA constructs including the *unc-54* 3' UTR were not able to restore normal pBoc/exp cycling, unlike constructs containing the *rab-3* 3'UTR (Fig 2D). pBoc cycles observed, L to R: 49, 119, 27, 25, 20, 18. Kolmogorov-Smirnov test was used. ns, not significant, * p < 0.05, ** p < 0.05, *** p < 0.0005, **** p < 0.0001, Fisher's Exact Test. Error bars represent SEM. Control: *oxIs12*. (B) Percent stacked bar graph for visual scoring of Aex phenotype rescue. Animals were randomized on plates and scored by phenotype, then genotyped. Animals were scored as normal (no gut distention, strong pBoc contraction with accompanying expulsion), constipated (severe posterior gut distention, weak pBoc with no expulsion), or slightly con (some possible gut distention, normal pBoc, weak expulsion). Fisher's Exact test was used. * p < 0.05, **** p < 0.0001. Control: *oxIs12*.
(TIF)

**S3 Fig. Intestinal re-expression of regeneration inhibitors rescues DMP defects.** Mutants in the *aex* pathway that inhibit regeneration also show defects in defecation, caused by a lack of waste expulsion (Exp) following posterior body wall muscle contraction (pBoc). D1 adult animals were randomly selected and observed for 10 DMP cycles, and the ratio of Exp/pBoc was plotted. Intestinal re-expression of *aex* genes involved in axon regeneration inhibition was sufficient to significantly restore pBoc/Exp ratio in all tested mutants, although Exp/pBoc rescue was not always complete. pBoc cycles observed, L to R: 40, 49, 50, 50, 56, 40, 56, 50, 49. 51. ns, not significant, ** p < 0.05, *** p < 0.0005, **** p < 0.0001, Fisher's Exact Test. Error bars represent SEM.
(TIF)

**S4 Fig. Two dense core vesicle tethering regulators do not affect axon regeneration.** (A) Relative axon length in control (*juIs76*) and *unc-31(e928)* mutants. Axons cut per genotype, L to R: 95, 79. Kolmogorov-Smirnov test was used. ns, not significant. (B) Relative axon length in control (*juIs76*) animals, and *hid-1* (*js722* and *js1058*) mutants. Axons cut per genotype, L to R: 57, 61, 40. Kolmogorov-Smirnov test was used. ns, not significant.
(TIF)

**S5 Fig. cab-1 and rab-27 show reduced fecundity.** One-day adult worms were placed onto empty NGM plates seeded with OP50 and left for 48 hours. Adults were removed and progeny counted. *rab-27* mutants show significantly decreased brood size compared to control animals, and *cab-1* mutants show more severe defects. The low brood size of *cab-1* mutants is not increased in *rab-27;cab-1* double mutants. Worms sampled, L to R: 9, 10, 7, 8. One-way ANOVA test was used. ns, not significant, ** p < 0.005, **** p < 0.0001. Error bars represent SEM.
(TIF)

**S1 Table. Generated *C. elegans* strains.**
(DOCX)

## Acknowledgments

We thank WormBase and the Caenorhabditis Genetics Center (CGC), which is funded by the National Institutes of Health (NIH) Office of Research Infrastructure Programs (P40

OD010440). We also thank Tyler Page and Erik Jorgensen for suggestions and feedback regarding *cab-1*.

## Author Contributions

**Conceptualization:** Alexander T. Lin-Moore, Marc Hammarlund.

**Data curation:** Alexander T. Lin-Moore, Marc Hammarlund.

**Formal analysis:** Alexander T. Lin-Moore, Motunrayo J. Oyeyemi.

**Funding acquisition:** Marc Hammarlund.

**Investigation:** Alexander T. Lin-Moore, Motunrayo J. Oyeyemi.

**Methodology:** Alexander T. Lin-Moore, Marc Hammarlund.

**Project administration:** Alexander T. Lin-Moore, Marc Hammarlund.

**Resources:** Alexander T. Lin-Moore, Marc Hammarlund.

**Supervision:** Marc Hammarlund.

**Validation:** Alexander T. Lin-Moore.

**Visualization:** Alexander T. Lin-Moore.

**Writing – original draft:** Alexander T. Lin-Moore.

**Writing – review & editing:** Alexander T. Lin-Moore, Marc Hammarlund.

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
