## [Decision Letter · Decision Letter 0]

19 Feb 2021

Dear Dr Hammarlund,

Thank you very much for submitting your Research Article entitled 'rab-27 acts in an intestinal secretory pathway to inhibit axon regeneration in C. elegans' to PLOS Genetics.

The manuscript was fully evaluated at the editorial level and by independent peer reviewers. The reviewers appreciated the attention to an important problem, but raised some substantial concerns about the current manuscript. Based on the reviews, we will not be able to accept this version of the manuscript, but we would be willing to review a much-revised version. We cannot, of course, promise publication at that time.

If you decide to revise the manuscript for further consideration at PLOS Genetics, please aim to resubmit within the next 60 days, unless it will take extra time to address the concerns of the reviewers, in which case we would appreciate an expected resubmission date by email to plosgenetics@plos.org.

[LINK]

We are sorry that we cannot be more positive about your manuscript at this stage. Please do not hesitate to contact us if you have any concerns or questions.

Yours sincerely,

Anne C. Hart

Associate Editor

PLOS Genetics

Gregory P. Copenhaver

Editor-in-Chief

PLOS Genetics

Reviewer's Responses to Questions

**Comments to the Authors:**

Reviewer #1: Uploaded Attached Document

Reviewer #2: In the manuscript, Lin-Moore and colleagues described the role of rab-27 in axonal regeneration, and they showed that rab-27 likely functioned in intestine to regulate the release of neuropeptides such as NLP-40 to inhibit axon regeneration upon injury. This is an interesting finding, but there are several questions should be carefully addressed to reach the conclusion.

Major concerns:

1. In a previous work, authors reported that rab-27 cell-autonomously inhibited axon regeneration. In this study, new evidence suggested that rab-27 might function in both intestine and GABAergic neurons to inhibit axon regeneration. This conclusion was reached by rescue experiments with different promoters and 3’utr. As discussed by the authors, there are many technique limitations for the rescue experiments using multiple copies transgenes. As this is the main conclusion of the manuscript, and it is somewhat inconsistent with the results from their published study, it is necessary to specifically knock out rab-27 in intestine and neurons (using Cre-Loxp or FLP-FRT system) to confirm its role in axon regeneration.

2. NLP-40 was identified as one of the peptides involved in regulating axon regeneration. As in the literature cited by authors, NLP-40 is expressed not only in intestine but also in coelomocytes (potentially others). Therefore, intestine specific recue of nlp-40(lf) is needed to reach the conclusion that nlp-40 is released from the intestine to affect axon regeneration. Similarly, intestine specific rescue of NPDC1/cab-1, SNAP25/aex-4, KPC3/aex-5 are also necessary to reach the conclusion. Also there wasn’t any evidence to show that RAB-27 inhibits axonal regeneration through NLP-40 secretion. To reach this conclusion, the authors at least need to carry out: 1) Double mutant analyses of rab-27;nlp-40; 2) label nlp-40 with GFP(or other tag) and show that axotomy can trigger the release of nlp-40 in a rab-27 dependent manner.

Minor:

1. Authors should check the data carefully. n number indicated in the figure legend are inconsistent with the dots number in some figures (for example in figure 1B).

2. n numbers are very different in each group. How is the n number determined in each group?

3. Why the difference of distribution probability of “relative axon length” from the same experiment group in different Figures are largely different? One example is the control group in figure 1B and in figure 4B.

4. please label the mean value in an easily distinguishable color.

Reviewer #3: The authors follow up on the previous finding that Rab-27 function inhibits axon regeneration. In this manuscript they convincingly demonstrate that Rab-27 functions in the intestine to inhibit regeneration, together with other neuropeptide secretion machinery. While the core of the paper is reasonably well-supported, the variability of the regeneration assay means that negative results are often over-interpreted and there are some general concerns about reproducibility. In addition, the story is sold as being evidence that axon inhibitory factors can act at a distance. For this point to really be made, major points 4 and 5 below would need to be addressed.

Major points.

1. The confirmatory experiments to show that the neuropeptide vesicle release pathway functions in intestinal cells are all done using global mutants. While the double mutant analysis suggests that intestinal function is likely, it would be cleaner to do some cell-specific manipulations in the intestine.

2. The Munc13b experiment is confusing- mutants are described as blocking secretion involved in the DMP motor program, but no effect was found here. Did the authors check to make sure that DMP is blocked in this genetic background? Similarly, were DMP-related phenotypes checked in hid-1 mutants as a control? And are they really convinced that these are negative results in terms of regeneration? In addition, the regeneration assay seems so variable that I think a little more caution may be warranted in the interpretation of these negative results (see below as well).

3. Is it the expulsion motor program that is required to inhibit regeneration? These neurons normally respond to signals from the intestine, so absence of signaling before injury may put these cells into a different program because they are not receiving correct input. Is there another way to block the circuit without disrupting peptide secretion to test whether it is really the secretion per se that is inhibiting regeneration, or whether an inappropriately functionin g circuit puts the cells in a pro-regenerative state before injury? Perhaps the aex-2 experiment addresses this? Is circuit function similarly compromised in this mutant background as in the others that improve regeneration? However, I am not convinced that aex-1 and aex-2 (Figure 3) do not improve regeneration. Their average is similar to aex-5, and just because they are not statistically significantly different from controls does not mean there is no effect. It can also mean that not enough animals were tested to detect an effect – which could well be the case because the assay is so variable.

4. How close in space are the intestine and GABA motor neurons? Is it possible to use existing EMs to show the spatial arrangement of these two cells types relative to one another?

5. Intestinal cells normally signal to GABA neurons to control defecation. I am therefore curious whether the influence of intestinal secretion on regeneration is specific to these neurons or acts more generally in cells that are not normally responding to the intestine.

6. Methods seem incomplete. No mention of when axon regeneration was scored, and several timepoints were used. Also, the data acquisition for Figure 2D is not described. And some of the methods seem to be in the figure legends beyond normal amounts- for example the specifics of injection plasmid concentration is in the legend to 2A.

Minor points.

In figure 3 why are some comparisons to oxls12 and some to juls76?

Sometimes controls are labeled control, sometimes oxls12. Are they all the same genotype? What is the actual genotype and why was it chosen? What are the different control datasets in different figures? They are quite variable with average axon length at 24h ranging from 0.5ish to 0.8. As some of the phenotypes are quite subtle against the backdrop of this intrinsic variability I have some overall concerns about reproducibility of the data. For example in Figure S3 the control data set seems to have much higher regeneration than other controls, and the aex-1 looks like it regenerates as well as many of the things that are called as having a phenotype in other figures.

Rabphilin mis-spelled in one place (as Rabhilin)

Why is the y axis labeled relative axon length in some figures and normalized axon length in others?

**Have all data underlying the figures and results presented in the manuscript been provided?**

Reviewer #1: Yes

Reviewer #2: None

Reviewer #3: Yes

PLOS authors have the option to publish the peer review history of their article (what does this mean?). If published, this will include your full peer review and any attached files.

Reviewer #1: No

Reviewer #2: No

Reviewer #3: No

---

## [Decision Letter · Decision Letter 1]

13 Oct 2021

Dear Dr Hammarlund,

We are pleased to inform you that your manuscript entitled "rab-27 acts in an intestinal pathway to inhibit axon regeneration in C. elegans." has been editorially accepted for publication in PLOS Genetics. Congratulations!

Yours sincerely,

Gregory P. Copenhaver

Editor-in-Chief

PLOS Genetics

Comments from the reviewers (if applicable):

Reviewer's Responses to Questions

**Comments to the Authors:**

Reviewer #2: The authors have addressed all my questions. Congratulations !

Reviewer #3: The authors have thoroughly addressed the reviewer comments with additional experiments, including double mutant analysis, and text changes.

**Have all data underlying the figures and results presented in the manuscript been provided?**

Reviewer #2: Yes

Reviewer #3: Yes

PLOS authors have the option to publish the peer review history of their article (what does this mean?). If published, this will include your full peer review and any attached files.

Reviewer #2: No

Reviewer #3: No

**Data Deposition**

http://datadryad.org/submit?journalID=pgenetics&manu=PGENETICS-D-21-00078R1

**Press Queries**

---

## [Editor Report · Acceptance letter]

10 Nov 2021

PGENETICS-D-21-00078R1 

rab-27 acts in an intestinal pathway to inhibit axon regeneration in C. elegans. 

Dear Dr Hammarlund, 

We are pleased to inform you that your manuscript entitled "rab-27 acts in an intestinal pathway to inhibit axon regeneration in C. elegans." has been formally accepted for publication in PLOS Genetics! Your manuscript is now with our production department and you will be notified of the publication date in due course.

With kind regards,

Katalin Szabo

PLOS Genetics

On behalf of:
